



# Meteor radar observations of polar mesospheric summer echoes over Svalbard

Joel P. Younger[1,2], Iain M. Reid[1,2], Chris L. Adami[1], Chris M. Hall[3], and Masaki Tsutsumi[4]

[1]ATRAD Pty Ltd, 20 Phillips St, Thebarton, SA, 5031, Australia
[2]School of Physical Sciences, University of Adelaide, Adelaide, SA, 5005, Australia
[3]Tromsø Geophysical Observatory, UiT - The Arctic University of Norway, 9037, Tromsø, Norway
[4]National Institute of Polar Research, 10-3, Midori-cho, Tachikawa-shi, Tokyo 190-8518, Japan

**Correspondence:** Joel Younger (jyounger@atrad.com.au)

**Abstract.** A 31 MHz meteor radar located in Svalbard has been used to observe polar mesospheric echoes (PMSE) during summer 2020. Data from 19 July was selected for detailed analysis, with a focus on extracting additional information to characterize the atmosphere in the PMSE region. The use of an all-sky meteor radar adds an additional use to data collected for meteor observations and enables the detection of PMSE layers across a wide field of view. Comparison with data from a 53.5

5    MHz narrow-beam MST radar shows good agreement in the morphology of the layer as detected between the two systems. Doppler spectra of PMSE layers reveal fine structure, including regions of enhanced return that move across the radar's field of view. The relationship between range and Doppler shift of off-zenith portions of the layer enable the estimation of wind speeds with high temporal resolution during PMSE conditions. Trials demonstrate good agreement between wind speeds obtained from PMSE Doppler spectra and those calculated from specular meteor trail radial velocities. Combined with the antenna polar

diagram of the radar, this same relationship was used to infer the aspect sensitivity of observed PMSE backscatter, yielding a mean backscatter angular width of $6.6 \pm 2.8°$. A comparison of underdense meteor radar echo decay times during and outside of PMSE conditions did not demonstrate a strong correlation between the presence of PMSE and shortened underdense meteor radar echo durations.

## 1 Introduction

Temperatures in the summer polar mesosphere can fall below the local sublimation point of water vapor, allowing ice crystals to form, particularly when other types of aerosols contribute as condensation nuclei. Larger ice crystals have long been observed as noctilucent clouds (NLC) at high latitudes (Leslie, 1885). Radar can detect these layers as what Hoppe et al. (1988) coined polar mesospheric summer echoes (PMSE) (see e.g. Cho and Röttger (1997), Rapp and Lübken (2004)). Polar mesospheric clouds (PMC) in general are of particular interest to atmospheric studies, as they can be a proxy for changes in climate and

the impact of solar activity on the middle atmosphere (Thomas, 1996; DeLand et al., 2006). Kirkwood et al. (2002) found that temperature perturbations from 5-day planetary waves may be responsible for the low temperatures necessary to facilitate PMSE.



Since the initial detection of PMSE with VHF radar reported by Czechowsky et al. (1979) at 53.5 MHz and Ecklund and Balsley (1981) at 50 MHz, there have been numerous radar studies of PMSE (see e.g. Hocking (2011)). Hoppe et al. (1988) found
PMSE detectable by a 224 MHz incoherent scatter radar, indicating that the Bragg scatter condition is satisfied over a wide range of physical scales. Klekociuk et al. (2008) conducted common-volume measurements in Antarctica of polar mesospheric clouds using lidar and PMSE using a 55 MHz MST radar, finding 70% overlap between the different sensors' detections of the two phenomena. Kaifler et al. (2011) presented a similar study in the northern hemisphere including a decade of lidar and MST radar observations of NLC and PMSE. Morris et al. (2004) and Morris et al. (2006) observed PMSE using an MST radar
in Antarctica, confirming that southern hemishpere PMSE has similar morphology to that seen in the northern hemisphere.

There has also been interest in observing PMSE using meteor radars. These systems are usually comprised of 6 antennas in total with a much smaller array footprint than the more sensitive narrow beam mesospheric-stratospheric-tropospheric (MST) radars. Typically used for the determination of winds and temperatures in the 80-100 km region, there is also the possibility of using them for PMSE study. Swarnalingam et al. (2009) used all-sky meteor radars in and around the arctic circle to estimate
the effective radar cross section of PMSE scatter. Most recently, Hall et al. (2020) provided an initial report of simultaneous detections of PMSE by the same narrow beam MST radar and all-sky meteor radar used in this study.

## 2 Radars

Data from two radars near Longyearbyen in the Svalbard archipelago (UTC +1) are used in this study to compare observations of PMSE return. An all-sky meteor radar is the primary instrument for exploring new methodologies and a narrow beam MST
system is used for complementary higher resolution measurements across a restricted field of view and as direct comparison between narrow beam and all-sky observations.

### 2.1 NSMR

The Nippon/Norwegian Svalbard Meteor Radar (NSMR) at 78.169°N, 15.994°E is a 31 MHz all-sky interferometric meteor radar transmitting with a peak power of 8kW. NSMR transmits a 4-bit complimentary code at a PRF of 430 Hz and samples
at a 1.8 km range resolution. Originally installed in 2001, it was upgraded in December 2019 to use the ATRAD Enahanced Meteor Radar (EMDR) transmitter and digital tranceiver (see e.g. Rao et al. (2014)).

NSMR uses a single circularly polarized three-element crossed Yagi transmit antenna (40° full-width at half maximum) and five receive antennas of the same design in a standard Jones cross (Jones et al., 1998). In this configuration, two perpendicular baselines of three antennas are spaced at 2 and 2.5 wavelengths from a shared central antenna. The angle of arrival of incident
scatter from meteor trails is determined by comparing the phase differences between different antenna pairs (Holdsworth, 2005).

Primary uses of all-sky meteor radar include using meteor detection radial velocities to infer wind speed and direction in the 80-100 km meteor region, as well as using the echo duration of underdense meteors to estimate the local ambipolar diffusion



coefficient and hence, temperature (Hocking, 1999; Cervera and Reid, 2000). More recently, meteor radar has also been used
to infer atmospheric density in the mesospher/lower thermosphere (Younger et al., 2015; Yi et al., 2018)

Analysis of return from PMSE was conducted using complex time-series records assembled from in-phase and quadrature
components measured for the received signal on each antenna. Reception channels for each antenna were added incoherently
to maintain the wide central beam pattern of the individual antennas. Meteor detection data were characterized using ATRAD
analysis software as described by Holdsworth et al. (2004).

## 2.2 SSR

The Svalbard SOUSY Radar (SSR) is a narrow-beam MST radar located at 78.170°N, 15.990° that transmits at 53.5 MHz
with a peak power of 8kW. SSR transmits a 16-bit complimentary code at a PRF of 1400 Hz and samples at a 0.5 km range
resolution. Based on the mobile SOUSY design (Czechowsky et al., 1984), it has undergone a number of upgrades and changes
to configuration (Zecha et al., 2001; Hall et al., 2009), most recently being the installation of a new transmitter and digital
transceiver of the same design as NSMR in April 2019. SSR currently uses 356 linearly polarized four-element Yagi antennas
to transmit a single 5° full-width at half maximum vertical beam.

Typically, observation time is split between mesospheric and tropospheric observations in one minute intervals. As with
NSMR, complex time-series data were searched for possible PMSE return. It should be noted that results from SSR data are
only shown in Fig. 2, with all other results being from NSMR unless otherwise specified.

## 3 PMSE detection

Following the initial investigation by Hall et al. (2020), NSMR and SSR data for 18-20 July, 2020 were analyzed to study
PMSE detections in more detail. Weak PMSE was intermittently detected on 18 July between 0600-1100 (all times UTC) and
between 0730-1130 on 20 July. In addition to the low PMSE signal strength and intermittent occurrence, both these detection
periods also displayed significant interference. PMSE was clearly detected on 19 July, including over two hours with large
signal strength. Data from 19 July is used for illustrative purposes throughout this paper.

Detection of PMSE by NSMR peaked in the 85.5-87.3 km range bin, across different times and with different strengths,
as can be seen in Fig. 1. At 0500-0530 a small PMSE-like feature was detected. One minute Doppler profiles (described in
section 3.2) for 0647-0738 showed sporadic detections of a very weak PMSE-like feature around 90 km that is not apparent
in the range-time intensity plot. A period of strong PMSE detection started at 0901 and continued until 1220, with several
sub-peaks. The intensity of the main PMSE detection gradually declined from about 1100, with a low-intensity period seen
until around 1220. PMSE detections by NSMR exhibited vertical smearing on the range-time intensity plot above the primary
detection range, which is due to off-zenith detection of the approximately constant height PMSE layer at greater ranges. Strong
ionospheric return was also detected by NSMR from 1410 to 2245 (not shown).


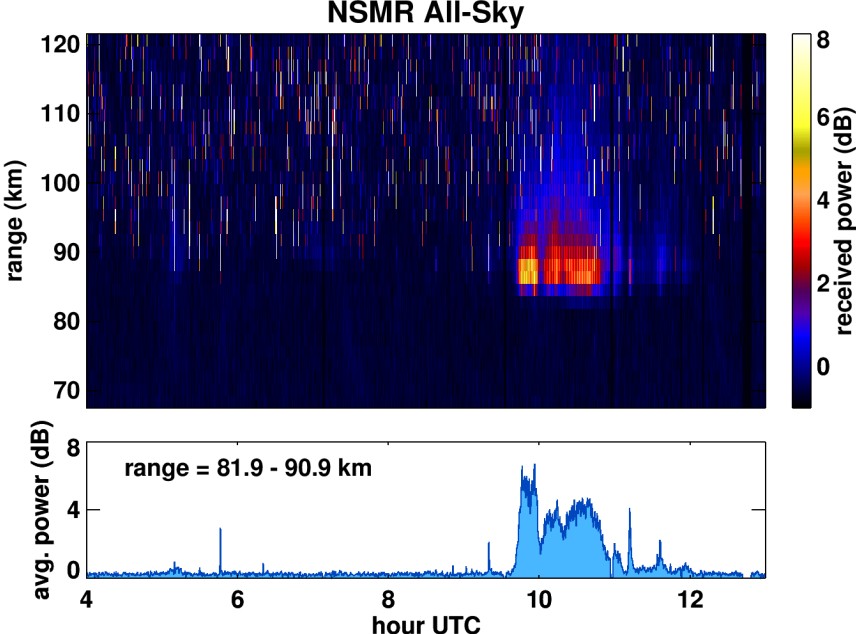

**Figure 1.** NSMR all-sky received power (incoherently averaged across all five antennas) for 19 July, 2020. 30 second averages in 1.8 km range bins. Plot intensity has been capped at 8 dB to enhance the visibility of weak features. The bright vertical segments above 85 km are meteor echoes.

### 3.1 Comparison of all-sky and narrow beam observations

The observations of PMSE by SSR's narrow vertical beam seen in Fig. 2 strongly correlate with the observations by NSMR. SSR detected the 0500-0530 PMSE-like feature more strongly than NSMR and displayed split layer behavior that was not seen in NSMR data. SSR's detection of the 0647-0738 layer was also much stronger than what was seen by NSMR, with two layers clearly visible on the range-time intensity plot. The main PMSE detection by SSR shares a similar time evolution to that seen by NSMR, with transient layer splitting detected between 83-88 km. SSR observations do however exhibit a more gradual

decrease from 1100-1200, as opposed to the decrease to a low SNR plateau seen by NSMR.

SSR has the advantage of having a 0.5 km range resolution, as opposed to 1.8 km for NSMR. Combined with the focusing of power into a narrower beam, this allows finer details in the PMSE layer to be seen, including split layers and dynamic upper and lower edges. One key difference to NSMR observations is the lack of vertical smearing above the layer, which supports the interpretation that the vertical smearing in NSMR's range-time intensity plot is due to off-zenith detection of a thin layer.

The split layers seen by SSR are consistent with higher resolution measurements produced by the MAARSY narrow beam VHF radar (Czechowsky et al., 1989) and the EISCAT VHF incoherent scatter radar (Röttger et al., 1988). The observations of Cho and Röttger (1997) in particular also show periods of split layer PMSE, in addition to periods of continuous return

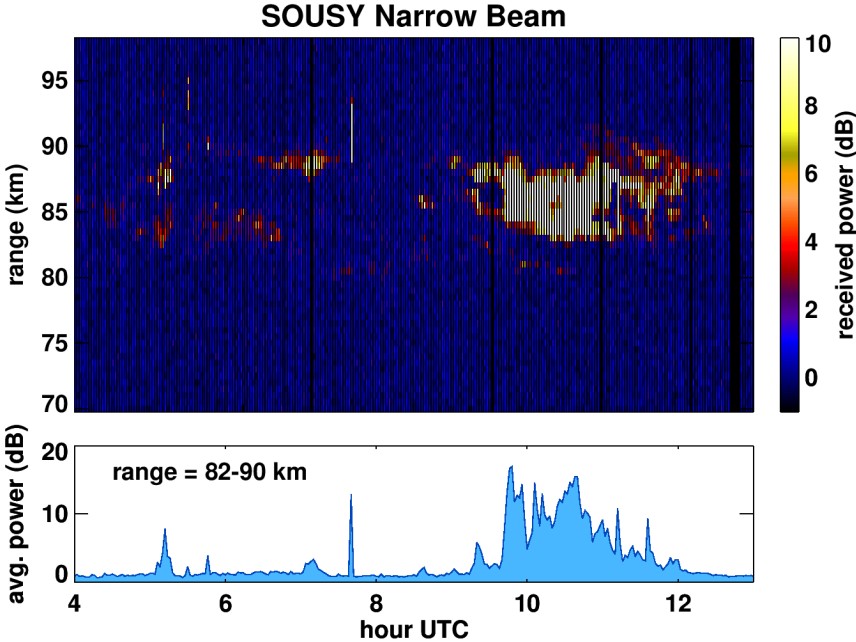

**Figure 2.** SSR narrow vertical beam received power for 19 July, 2020. One minute averages in 0.5 km range bins. Plot intensity has been capped at 10 dB to enhance the visibility of weak features. Vertical striping is due to interleaving mesospheric observations with other experiments at one minute intervals.

across the entire PMSE region. The presence of split PMSE/PMC layers may be further evidence of complex mesopause structures, with multiple distinct local temperature minima (She and Von Zahn, 1998; Thulasiraman and Nee, 2002) allowing

for the formation of PMC at multiple heights.

### 3.2 Meteor radar PMSE Doppler profiles

The observation of PMSE layers by a wide field-of-view radar has the advantage that different portions of the horizontal extent of the layer may be detected at differing ranges and Doppler shifts. The curvature of the range-Doppler profile of PMSE detection is related to the speed of the background wind with which the layer is moving. For each Doppler frequency component

the minimum detected range corresponds to return from along the zenith-wind vector plane. Figure 3 shows several examples of Doppler profiles associated with PMSE layers. PMSE mostly presents in the Doppler profiles as arcs curving upward from the 0-Doppler detection of the layer, the point which corresponds to return from around zenith.

The first two profiles from 0506 and 0514 are from the weak transient PMSE layer detected by NSMR and SSR. These two profiles differ from the profiles seen for the main detection period in that they exhibit a pronounced asymmetry and an almost

110 linear range-Doppler relation. This may indicate that the scattering geometry for the early transient PMSE detection may differ from that of the main PMSE detection.

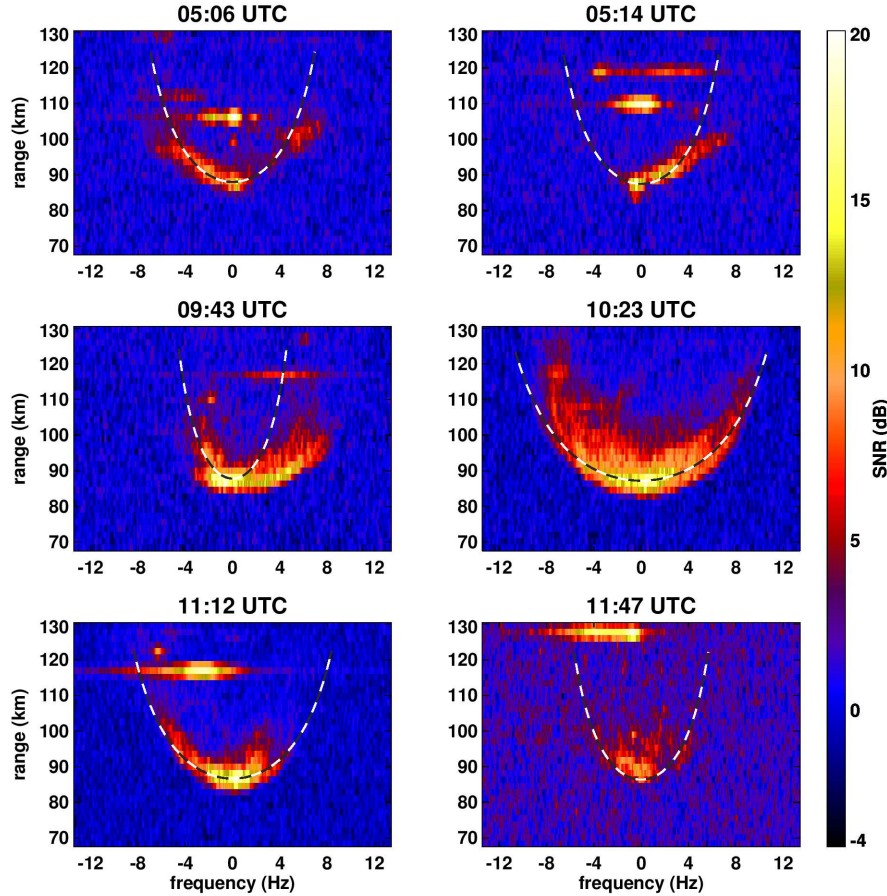

**Figure 3.** NSMR range-Doppler profiles for 19 July, 2020, constructed from one minute observation periods. Top left: transient PMSE detection. Top right: Asymmetric transient PMSE. Middle left: Asymmetric onset of strong PMSE return. Middle right: strong PMSE return exhibiting fine structure. Bottom left: end of strong PMSE period displaying asymmetric intensity distribution. Bottom right: End of PMSE detection period primarily around 0-Doppler. Dashed lines show the expected range-Doppler profile of a thin layer, based on speed calculated from meteor trail detection radial velocities.

The 0943 profile displays an asymmetric Doppler profile at the onset of strong PMSE return. This is indicative of an anisotropic wind field, as the layer is seen as a region of slow winds (more vertical, negative portion of the profile) which is being replaced as wind speed increases above the radar. At 1023 there is strong detection during the main PMSE layer period, with fine structure apparent including layer splitting visible near the edges and multiple small, persistent features. The profile for 1112 shows the main layer detection as it decreases in amplitude. The mostly negative Doppler asymmetric profile



is consistent with the scattering layer leaving the radar's field of view. At 1147, PMSE SNR is decreasing towards the end of the detection period and significant SNR is limited to around 0-Doppler.

### 3.3 Estimating wind speed from range-Doppler profiles

If it is assumed that PMSE occurs in a thin layer of approximately constant height, then range-Doppler profiles can be used to estimate the wind speed in the PMSE region.

The range $R$ to a point at zenith angle $\theta$ and height $h$ above the approximately spherical surface of Earth is given by

$$R = \frac{R_\oplus + h}{\sin\theta} \sin\left[\theta - \sin^{-1}\left(R_\oplus \sin\frac{\sin\theta}{R_\oplus + h}\right)\right] \tag{1}$$

where $R_\oplus$ is Earth's local radius. Here, the oblateness of Earth is neglected, which is justified on the ground that PMSE is
125 detected primarily at zenith angles within $\pm30°$ at around 86.4 km. This means that the horizontal extent of PMSE detected by meteor radar is not more than 100 km, so there will not be variation in $R_\oplus$ sufficient to significantly affect equation 1.

The basic radar Doppler equation for a radar transmitting at frequency $f_0$ and wind speed $V$ with radial component $v_r$

$$\Delta f = \frac{2f_0 v_r}{c - v_r} \approx \frac{2f_0 v_r}{c} \tag{2}$$

can be rearranged, assuming a horizontal wind, to infer the zenith angle of a component of the spectrum with Doppler shift
$\Delta f$, as

$$\theta = \sin^{-1}\left(\frac{c\Delta f}{2f_0 V}\right), \tag{3}$$

where $c$ is the speed of light.

Considering the return from the zenith-wind vector plane, it will form a distinct bottom edge to the range-Doppler profile of PMSE return. Therefore, the angular dependence of equation 3 can be used to describe the relationship between zenith angle
and Doppler shift in the zenith-wind vector plane. Inserting then equation 3 into equation 1, we have a function $R(\Delta f, h, V)$, that specifies the curve of the range-Doppler profile of a scattering layer moving horizontally at height $h$ with speed $V$, resulting in Doppler shifts of $\Delta f$.

For NSMR range-Doppler profiles, a least squares fit was applied to determine the wind speed parameter of $R(\Delta f, h, V)$ for observation periods where the peak PMSE layer SNR was at least 6 dB. Overall, PMSE-based estimates of wind speed,
assuming $V$ is purely horizontal, were mostly in keeping with estimates obtained from the more conventional meteor trail radial velocity technique. This comparison is discussed in further detail in section 4.2.

### 3.4 PMSE Doppler profile sub-structures

Beyond the range-Doppler relationship due to background winds, the spectra in Fig. 3 also display smaller scale return that are indicative of scattering from sub-structures within the PMSE layer. In some cases, it can also be seen that the background wind
moves regions of enhanced scatter through the field of view of the radar.

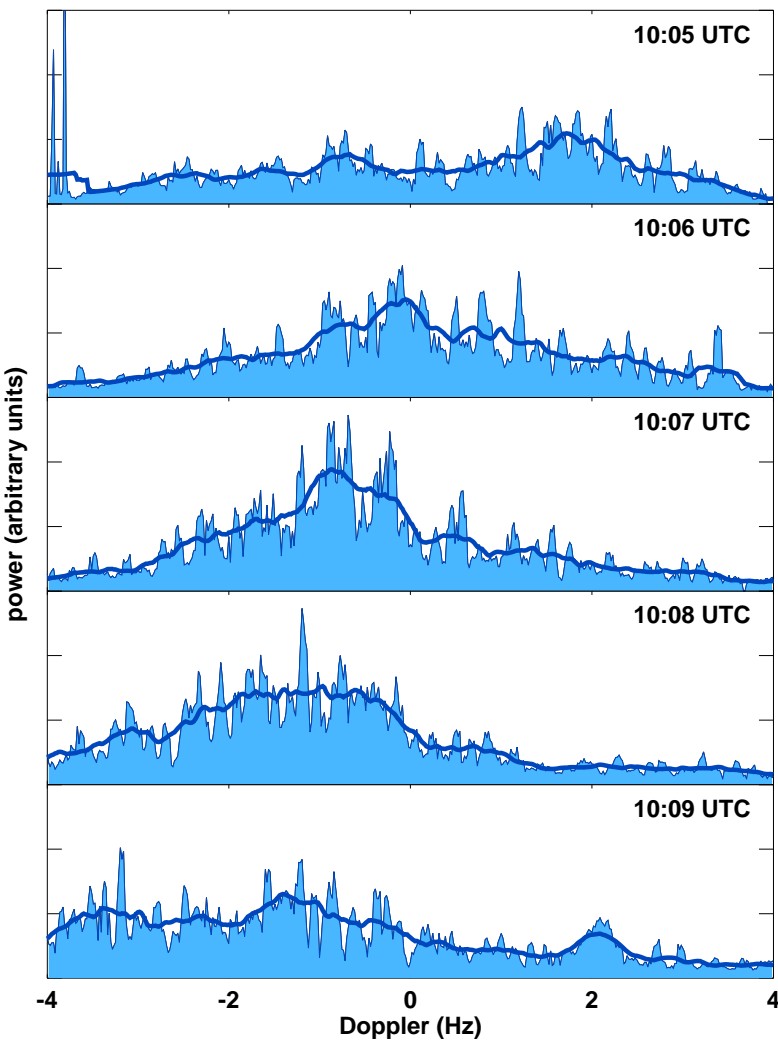

**Figure 4.** Movement of a perturbation in profiles of spectral peak power of PMSE detection for NSMR on 19 July, 2020. Solid line is spectral power smoothed with a 0.5 Hz window.

Figure 4 shows the peak power for one minute PMSE range-Doppler spectra. In this series of plots, it can be seen that a region of enhanced signal return moves from positive to negative Doppler. Furthermore, a cursory examination reveals that the speed at which the enhanced return travels from positive to negative Doppler is consistent with the background wind derived



from meteor detections and fits to the range-Doppler profile. This demonstrates that the PMSE layer is not homogeneous, but
contains moving regions of enhanced reflectivity that alter the shape of the spectral power distribution over time.

## 4    Comparison with meteor detections

The use of a meteor radar to observe PMSE also presents the opportunity to use conventional meteor radar detections to provide
additional information about the state of the atmosphere during and around PMSE detection periods. Meteor radar detections
are commonly used to estimate winds in the 80-100 km height range. PMSE derived wind speeds provide an opportunity
to verify the accuracy of meteor derived wind estimates. Furthermore, PMSE has been implicated in the anomalously short
decay times of underdense meteor echoes below 90 km. The direct detection of PMSE by meteor radar simplifies the process of
assessing the effect of PMSE on meteor echo decay times, which also relates to the broader study of middle atmosphere plasma
chemistry (see e.g. Rapp and Lübken (2001), Murray and Plane (2003), Murray and Plane (2005), Friedrich et al. (2011)).

### 4.1    Meteor winds

Winds were estimated in a conventional manner using meteor detection radial velocities to produce wind profiles with 30
minute and 2 km vertical resolutions. Meteor-based winds were calculated (assuming the vertical wind $w = 0$) using a least
squares fit to the relation

$$v_r = ul + vm \tag{4}$$

where $v_r$ is the radial velocity of the meteor trail, $u$ and $v$ are the zonal and meridional components, and $l$ and $m$ are the direction
cosines (see e.g. Holdsworth et al. (2004)). Prior to wind estimation, the observed zenith angles $\theta$ of meteor detections were
converted to local zenith $\theta_{loc}$ angles using the relation

$$\theta_{loc} = \sin^{-1}\left(\frac{\theta}{R_\oplus + h}\right). \tag{5}$$

$R_\oplus$ was calculated at NSMR's latitude using the WGS84 ellipsoid (Decker, 1986).

Outlier rejection was implemented by checking the predicted $v_r$ for each meteor and rejecting any detections differing by
170 more than 30 m s$^{-1}$. Wind components were then recalculated with the remaining meteors and the process was repeated until
all predicted $v_r$ were within tolerance. If less than 6 meteors were present in the height/time bin or were left after outlier
rejection, it was considered an empty bin.

Seen in Fig. 5, the meteor wind profiles show that the main PMSE detection from 0901-1220 coincides with the semi-diurnal
tide maximizing the eastward wind just above the layer height and the northward meridional wind maximizing around the layer
height.

Vertical wind shear was also calculated as the magnitude of the vector difference between winds in adjacent height bins.
The main PMSE detection period at 0901-1220 occurred during moderate vertical shear, but an examination of the relationship
between shear conditions and the occurrence of earlier transient PMSE layers was inconclusive.





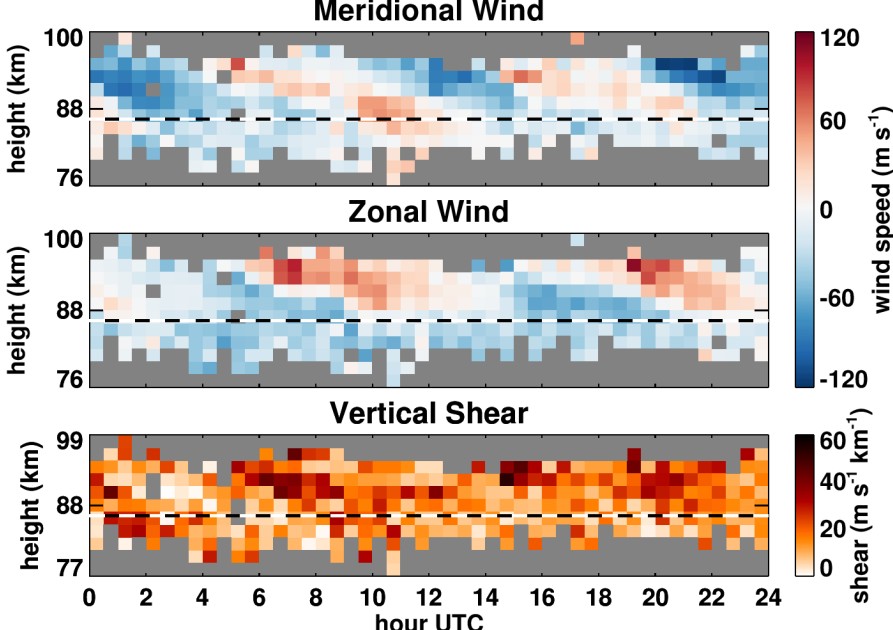

**Figure 5.** 30 minute averages of zonal (top) and meridional (middle) components of wind and vertical wind shear magnitude (bottom) calculated from NSMR meteor detection radial velocities in 2 km height bins. The dashed line shows the approximate height (range bin of maximum intensity) of PMSE detection by NSMR. Gray squares denote insufficient data.

## 4.2 Comparison of meteor and PMSE Doppler winds

In order to compare the observed Doppler profiles of PMSE detections with local wind conditions, winds were estimated using meteor detections for each range-Doppler profile. The wind in the layer region was estimated for each PMSE profile using meteors detected within ±15 minutes of the profile time and within ±1 km height of the layer's 0-Doppler maximum intensity range.

Seen as dashed lines in Fig. 3, the meteor wind estimates closely match the peak power of the range-Doppler profiles of
PMSE return. This is consistent with the interpretation that observed scatter from PMSE seen by NSMR is from a thin layer as seen across a wide field-of-view. The asymmetric range-Doppler profile for 0943 shows good agreement for the negative Doppler portion of the spectrum, but not the positive Doppler, which is again consistent with a changing wind field in the radar's field of view.

When wind speed estimates from PMSE Doppler and meteor trail radial velocities are directly compared, as shown in Fig.
6, it is seen that the range-Doppler estimates of horizontal wind at the height of maximum PMSE return power are mostly in good agreement with the estimates obtained from meteor trail radial velocities during the main PMSE detection period.





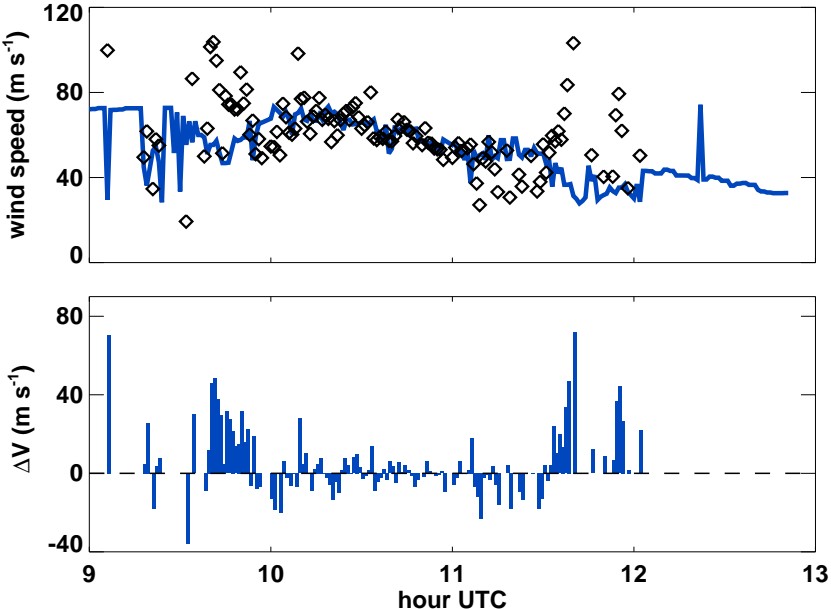

**Figure 6.** Top panel: Horizontal wind estimates made using one minute PMSE range-Doppler fitting (diamonds) and a moving 30-minute window of meteor trail detections (solid line). Bottom panel: Deviation between meteor and PMSE range-Doppler wind estimates.

Meteor and range-Doppler estimates of horizontal wind speed do not, however, agree at the beginning and end of the primary PMSE detection period for differing reasons at each time. At the beginning of the PMSE detection period, the wind field exhibits significant anisotropy, as evidenced by the asymmetry of the range-Doppler profile at 0943 in Fig. 3. During this
time, horizontal wind speed is increasing with the semi-diurnal tide, resulting in a divergent wind field with incoming high speed (positive Doppler) winds impinging on a region of slower winds leaving the radar's field of view (negative Doppler). It should however be noted that at NSMR's latitude of 78.169° N, the semidiurnal tide's zonal wavelength is approximately 4095 km. Compared with the horizontal extent of detected PMSE of about 100 km, this indicates that the observed Doppler asymmetry is not strictly tidal in nature, but more likely due to local transient features of the wind field.
A similar reversed situation, albeit with a smaller effect, is seen in the 1112 example, where the high speed region of the wind field is departing with the incident positive Doppler component displaying a noticeably smaller Doppler. It is also possible that the negative excursion around 0925 in the meteor wind estimate is due to the same causal factor as the similar negative excursion in the range-Doppler wind speed estimate approximately ten minutes later. In this case, it is useful to point out that meteor detections occur across a substantially larger field of view than PMSE, encompassing a radius of approximately 300
205 km, as opposed to an approximately 50 km maximum radius for the detected horizontal extent of PMSE.

As wind speed estimates based on meteor trail radial velocities are dependent on the distribution of meteors within the radar's field of view, there can be times when meteor detections are concentrated more in some parts of the field of view than





others. In an anisotropic wind field, this may lead to excess weight being placed on regions of the sky where meteors happen to be detected for a particular observation period. When comparing wind estimates made from the range-Doppler profiles of a
thin PMSE scattering layer in a smaller region of the sky with wind estimates made from meteor detections scattered across a larger area, it may be that wind speed estimate perturbations seen by the different methods may be the result of sampling different regions of a divergent wind field.

The disagreement between meteor and range Doppler wind speed estimates at the end of the primary PMSE detection period is due to a different mechanism. From about 1140-1200, the significant PMSE SNR in the range-Doppler profile is limited to
a narrow spectral region around the 0-Doppler component. Under this condition, the applied fit does not produce an accurate range-Doppler curve. The result is an erroneously flat fit, which corresponds to an overestimate of wind speed. It should be noted that the narrow, flat return at 0-Doppler is also indicative of a more aspect sensitive scatter mechanism, wherein detected backscatter is only visible near zenith.

### 4.3   Meteor echo decay times

Underdense meteor trails, with linear electron densities of less than $2.4 \times 10^{14}$ electrons m$^{-1}$ (McKinley, 1961), produce radar echoes that decay at an exponential rate governed by the local ambipolar diffusion coefficient $D$ (Lovell et al., 1947). The time, $\tau$, for an underdense meteor trail's radar echo to decay to a factor of $e^{-1}$ of the initial maximum is given by

$$\tau = \frac{\lambda^2}{16\pi^2 D} \tag{6}$$

where $\lambda$ is the frequency of the radar. This relation is the basis of methods to estimate temperature in the meteor ablation region
either by using the slope of $\log \tau$ as a function of height (Hocking, 1999) or by supplying pressures to the relation

$$D = 6.39 \times 10^{-2} K_0 \frac{T^2}{p} = 2.23 \times 10^{-4} K_0 \frac{T}{\rho}, \tag{7}$$

where $K_0$ is the zero field mobility of the diffusing ions (Mason and McDaniel, 1988), and $T$, $p$, and $\rho$ are the atmospheric temperature, pressure, and density, respectively (Cervera and Reid, 2000).

It should be noted that this relation only holds for the case where only ambipolar diffusion is responsible for the evolution of
meteor trail plasma. It has been observed that meteors detected at lower altitudes, especially below 85 km, have significantly shorter decay times than is predicted by diffusion alone (Kim et al., 2010). Lee et al. (2013) and Younger et al. (2014) showed that this is most likely due to the neutralization of meteoric plasma initiated by the attachment of free electrons to neutral $O_2$ and $N_2$ in a three-body process. It is possible that the ice crystals thought to be responsible for PMSE also affect the observed decay time of meteor trail echoes, as electrons can attach to ice crystals, leading to additional crystal growth and meteoric
plasma neutralization. If this mechanism plays a significant role on meteor trail evolution, then meteor trail decay times should differ in the presence of PMSE.

The meteor trail echo decay times seen in Fig. 7 show some correlation between anomalous decay times and PMSE occurrence as minor negative excursions to decay time. The lack of a more dramatic correlation could be due to the dominance of neutral three-body attachment removing free electrons from the trails, as compared to the removal rate due to aerosol at-





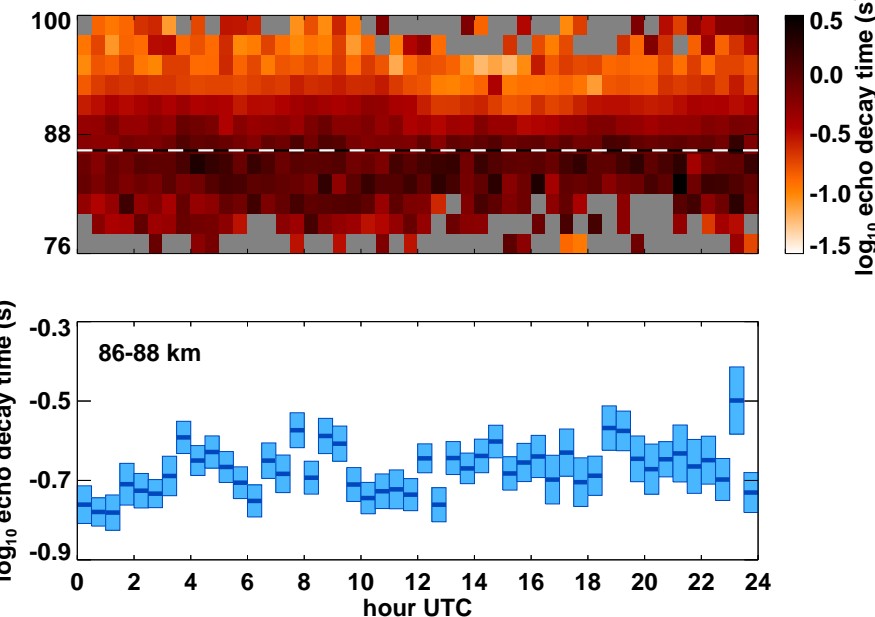

**Figure 7.** 30 minute averages of echo decay times of underdense meteors detected by NSMR in 2 km bins. The dashed line in the upper panel shows the height of the range bin that PMSE return is maximum. Bottom panel shows the 30 minute averaged decay time of meteors around the PMSE height. Shaded boxes denote 95% confidence interval.

tachment to PMC ice crystals. The small negative excursions in decay time coincident with PMSE around 0530-0630 and 0930-1300 may be consistent with the findings of Laskar et al. (2019), who estimated an approximately 10% decrease in meteor decay times in the presence of PMC, although more work is needed to determine if the variation observed by NSMR is due to PMC effects or geophysical variability.

     It should be noted that Laskar et al. (2019)'s use of NLC occurence differs from our use of PMSE in that PMC/NLC ice
crystals are thought to be larger and concentrated at the lower edge of the PMSE region. An examination of NSMR data showed that meteor decay times in lower height bins displayed more temporal stability than meteor detections in the 86-88 km height bin, which suggests that distortion of meteor decay times is not significant at the lower edge of the detected PMSE region. Furthermore, previous work has indicated that the presence of PMC may actually slow the neutralization of meteor trails by the depletion of mesospheric atomic oxygen (Murray and Plane, 2003). Whatever the precise details of the interaction
between PMC particles and meteoric plasma, the presence of detected PMSE cannot conclusively be proven or ruled out as the primary causal factor in reducing meteor radar echo decay times in this case. An examination of NSMR data across all seasons including a cross-comparison with PMSE detection and non-detection periods is required to definitively answer the question with appropriate statistical rigour.



## 5 Aspect sensitivity

The detection of Doppler components of the PMSE layer away from zenith present an opportunity to estimate the angular dependence of observed backscatter from PMSE. There are however some limitations that the large beamwidth of NSMR imposes on attempts to infer the aspect sensitivity of observed PMSE. The narrow beam expression for the aspect sensitivity parameter $\theta_s$ as in Hocking et al. (1986) is not applicable in this case, as using equation 3 with range-Doppler profiles allows us to directly sample received power within the beam at different zenith angles, rather than tilting the beam. Similarly, the sparse, widely spaced interferometer array makes the use of the Capon method (Sommer et al., 2014) impractical due to the complex beam pattern of the cumulative array. Furthermore, the wide central beam angle of the individual antennas is too large in comparison the diffraction pattern of individual scatterers to apply spatial correlation analysis (SCA) as in Sommer et al. (2016).

The significant Doppler information available does, however, present an opportunity to gain at least a qualitative description of the aspect sensitivity of PMSE. As described in section 3.3, the zenith angle of return from a thin layer in the zenith-wind vector plane can be converted to Doppler frequency and *vice versa*. The return from PMSE shown in the range-Doppler profiles of Fig. 3 presents as an arc with a partially filled interior. While return from regions away from the zenith-wind vector plane fills in the interior of the arc, the lower edge of the PMSE return arc corresponds to scatter from within the zenith-wind vector plane.

Hence, the peak powers observed in each frequency bin, which are in good agreement with the lower range boundary of PMSE return, provides an opportunity to translate observed Doppler shift into an estimate of zenith angle along the wind vector. The peak power at each zenith angle can then be used to infer the angular dependence of PMSE backscatter strength.

To do this for each profile, the PMSE range-Doppler estimate of wind speed was applied to equation 3 to produce an estimate of zenith angle. The peak power in each zenith (frequency) bin was estimated from the amplitude of a Gaussian curve fitted to power in the bin as a function of range. A Gaussian distribution was then fit to the peak powers of PMSE Doppler as a function of estimated zenith angle, corrected for antenna gain. The width of the fitted Gaussian curve is the PMSE aspect sensitivity parameter, $\theta_s$, and the center of the fitted curve is the offset from zenith or tilt angle.

In order to minimize contamination from meteor echoes, zenith-peak power profiles were limited those with maximum average power less than 600 (arbitrary hardware units). Profiles were also required to have successful Gaussian range/power fits with peak Doppler SNR between 3 and 30 dB in at least $40\%$ of zenith (frequency) bins. Finally, only Doppler bins in the frequency range of -2.5 to 2.5 Hz were used to exclude the majority of meteor detections that occur with higher Doppler values closer to the horizon.

Applying this process, $\theta_s$ was successfully estimated for 76 of the one minute observation periods between 0900-1300. The fitting process additionally provided the offset from vertical, which gives some indication of the preferential scattering or tilt angle of the observed PMSE. Seen in Fig. 8, $\theta_s = 6.6 \pm 2.8°$. The estimated aspect sensitivity showed considerable variation throughout the primary PMSE detection period. The offset of the zenith angle was close to zero with predominantly negative excursions, indicating that the observed PMSE scattered preferentially in the negative Doppler direction.



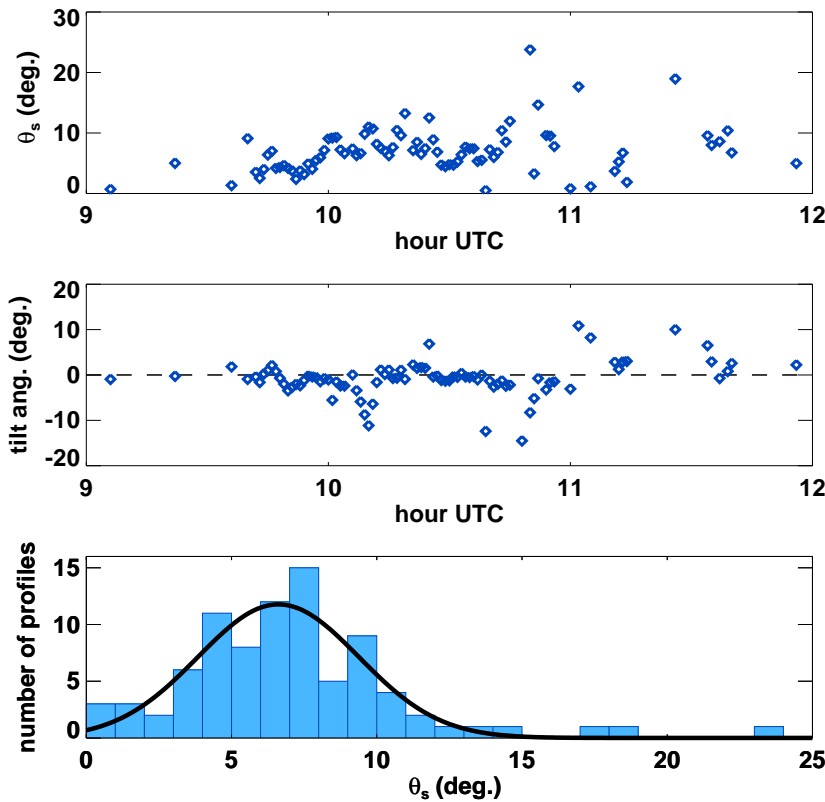

**Figure 8.** Aspect sensitivity of PMSE in the zenith-wind vector plane as observed by NSMR. Top panel: Aspect sensitivity $\theta_s$ of PMSE return obtained from Gaussian curve fitted to PMSE Doppler power as a function of Doppler estimated zenith angle. Middle panel: Center of Gaussian fitted to PMSE return peak power as a function of Doppler estimated zenith angle. Bottom panel: Histogram of PMSE backscatter $\theta_s$ estimates. Gaussian fit to the distribution of estimates shown as a solid line.

The mean and range of estimated aspect sensitivity values seen in Fig. 8 are consistent with other studies (see e.g. Reid (1990)). For comparison, Czechowsky et al. (1988), exploiting the sidelobes of a radar with similar configuration to SSR at Andenes, found values of 2-10° with typical values in the range of 5-6°. Swarnalingam et al. (2011) found a median value of 8-11° using a 51.5 MHz MST radar, with significant dependence on the height of the scattering layer. Larger values were estimated at higher altitudes, which is indicative of increasing isotropy with height. Smirnova et al. (2012), using a 52 MHz MST radar, found two populations of scatterers with aspect sensitivities of 2.9-3.7° and 9-11°, also showing an increase with altitude. Both these studies yielded similar results to the earlier work by Huaman and Balsley (1998) that gave mean values of 10° at 80 km and 14° at 90 km, but with substantial differences between radars at Andenes (5-6°) and Poker Flat (12-13°).



This study did not show a clear correlation between layer height and aspect sensitivity. However, it should be noted that the method used is only applicable to the height of maximum scattering intensity, so does not capture the full behavior of aspect sensitivity in different parts of the PMSE layer.

## 6  Conclusions

This study demonstrates that all-sky radars provide a useful complement to the more common narrow-beam studies of PMSE. The key advantage of all-sky systems is that they are able to capture Doppler contributions from PMSE continuously across a wide range of zenith angles. This reveals fine structure in PMSE layers and provides an immediate opportunity to infer the motion of the scatterers. The use of a 31 MHz radar is also noteworthy, given that most previous radar observations of PMSE have been conducted with MST radars with transmission frequencies above 50 MHz. This indicates that the $\lambda/2$ scattering condition is also fulfilled at larger spatial scales than for the more common 50 MHz and above observations. Thus, it has also been shown that the longer wavelength, which is optomized for meteor trail detection, is not a significant impediment to the detection of PMSE layers.

In particular, the range-Doppler profile of thin layer return obtained by wide field-of-view radars can be used to infer wind speed in the layer and the aspect sensitivity of the layer's scattering mechanism. A comparison of wind speeds obtained through this method and more conventional meteor echo based wind estimates shows good agreement for fully developed PMSE, an assessment that is also supported by the apparent motion of density perturbations within the distribution of received power from the layer. Aspect sensitivity estimated using range-Doppler profiles is consistent with previous estimates made using 51-52 MHz narrow-beam MST radars.

While this study was necessarily limited in its scope, the methods presented should in future be applied to longer data sets. Ideally, this will take the form of a campaign over summer at a polar location where frequent PMSE is observed. Additional data, such as lidar temperatures could also facilitate a more thorough interpretation of the results of the methods described.

*Data availability.* NSMR meteor detection data is available from http://radars.uit.no/MWR/NTMR/yyyymmdd_met.met where yyyymmdd is the date. Processed data and NSMR Doppler profiles are included in the supplementary data. Raw time series data is available upon request from The Arctic University of Norway.

*Author contributions.* JY developed the methodology, performed the analysis, and prepared the manuscript. IR and CA assisted with the investigation. CA advised on technical details of instrumentation. CH and MT manage NSMR. CH and CA manage SSR.

*Competing interests.* The authors declare that they have no conflict of interest.



*Acknowledgements.* This research was made possible by generous financial contributions from ATRAD Pty Ltd and employed data from instruments supported by the Research Council of Norway under the project Svalbard Integrated Arctic Earth Observing System—Infrastructure
development of the Norwegian node (SIOSInfraNor, Project No. 269927). The authors would like to thank the Arctic University of Norway and the National Institute for Polar Research of Japan for use of data from the Nippon/Norwegian Svalbard Meteor Radar and Svalbard SOUSY Radar.



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
