# Peer review of "Meteor radar observations of polar mesospheric summer echoes over Svalbard"

_Atmospheric Measurement Techniques, 2021_

## Author Response (AR1)

The authors would like to thank both reviewers for their consideration of the manuscript and helpful suggestions. Please note that line numbers and figure numbers in this response refer to the original manuscript, but figure numbering has changed in the revised manuscript. In addition to the responses to the reviewers' comments, the estimate of aspect sensitivity has been updated from 6.6 ±2.8° to 6.8 ±3.3°, following the discovery of an error in the antenna beam pattern calculation. Figure 9 has been updated accordingly. Responses (bold) to the reviewers' specific comments are listed below.

Response to reviewer 1:

Fig.1, Fig.2: I recommend showing the same height range in both figures.

**Changed range of figure 2 to match figure 1**

P4 L96: The given reference to Chechowsky et al., 1989 could not be related to MAARSY results.

**Changed "MAARSY" to "mobile SOUSY"**

P5 L104: The wide field-of-view is often mentioned here and elsewhere in the text and probably refers to the characteristics of the individual antennas. It would be interesting to find here some statements about the characteristics of the antenna pattern resulting from the five incoherently added receiving channels, which would clarify the mentioned advantage that different parts of the horizontally extended PMSE can be detected.

**Added to section 2.1: "While coherent addition of antennas would enhance sensitivity around zenith, the complexity of the sidelobe structure (see e.g. figure 1 of Chau, 2019), makes this unsuitable for the analysis of PMSE Doppler described in section 3.2."**

**Moved meteor analysis description to following paragraph and added:**

**"Meteor characteristics recorded include range, direction (angle of arrival), radial velocity, echo power, SNR, and echo duration."**

Fig.4: To what height do these spectra belong?

**Changed line 146 to read: "Fig. 4 shows the power at the strongest detection range in each frequency bin for one minute PMSE range-Doppler spectra.."**

**Changed Fig. 4 caption to read: "Movement of a perturbation in profiles of the power at the strongest detection range in each frequency bin of PMSE detection for NSMR on 19 July, 2020. Solid line is spectral power smoothed with a 0.5 Hz window."**

P10 L184: Fig.3 to Fig.5

**This is correct as written. For clarity, the sentence is changed to read "Seen as dashed lines in Fig. 3, the range-Doppler curves calculated from meteor wind estimates closely match the peak power of the range-Doppler profiles of PMSE return."**

P10 L184ff: The horizontal wind shown in Fig.5 is in the range 76–100km. The statement that "the dashed lines in Fig.3" show that "the meteor wind estimates closely match the peak power of the range-Doppler profiles of PMSE return" is therefore somewhat misleading.

**See response to above comment. We do not feel that the statement is misleading, but may have suffered from a lack of clarity. Changed sentence to read: "Seen as dashed lines in Fig. 3, the range-Doppler curves calculated from meteor wind estimates closely match the peak power of the range-Doppler profiles of PMSE return, as seen by the overlap between the dashed lines and PMSE intensity."**

Response to reviewer 2

The range resolution of the NSMR is mentioned as 1.8 km. Is this based on the transmitted pulse width or receiver range gate? This information will be important to mention. It is possible to oversample with lesser range gate times while the transmitter pulses are actually longer.

**Changed second sentence of section 2.1 to read: "NSMR transmits a 3.6 km long 4-bit complimentary coded pulses at a PRF of 430 Hz and samples at a 1.8 km range resolution."**

Being a paper on new technique, it will be helpful to add a schematic of the transmit receiver configuration of the radar (though this is commonly used formation for meteor wind radars, not all readers would have worked with them).

**Added an additional figure depicting the receive interferometer layout and individual antenna beam pattern. Added more precise description of the antenna beam FWHM to second paragraph of section 2.1.**

In section 3, it is mentioned that the investigation used NSMR and SSR data for 18-20 July 2020. However, no mention was there about the PMSE strength in SSR on 18 and 20 July. Is it weaker on those adjacent days in SSR? Including a range-time-intensity figure for the three days duration from both the radars will be valuable to get a rough idea how often such meteor measurements can be used.

**PMSE detection on 18 and 20 July is described in the first paragraph of section 3. We do not feel that three days of data can be used to give a rough idea of how often meteor radar can be used to study PMSE, but a campaign is under way to collect data over an entire summer season to answer this question.**

Figure 3 Doppler profiles. What are the intense horizontal echoes in addition to the U shaped Dopplers of PMSE?

**These are meteor echoes. Added to end of first paragraph of section 3.2: "Some contamination from meteor echoes is visible in the form of horizontal segments of return in the range-Doppler profile." Removal of meteor echo contamination from PMSE spectra will be described in an upcoming paper currently in work.**

Figure 3. Which altitude is attributed to the PMSE layer? The layer is thick in 0 Doppler frequency. Is the lowest point taken as PMSE altitude or the peak power in the 0 Doppler is taken as the altitude?

**Added to first paragraph of section 3.2: "The range of PMSE return on the range-Doppler profile at 0 Hz is the height of the layer."**

Line 125. PMSE is said to have observed within 30 deg around 86.4 km. In section 2 FWHM of the transmitted beam is mentioned as 40 deg. So how this 30 deg value is assumed or obtained? Whether this is applicable only at 86.4 km or to other PMSE altitudes too?

**Added to line 125: "(based on zenith angles calculated from equation 3 and the extent of observed PMSE Doppler)" Also changed description of NSMR beam to FWHM of 81.4-83.6°, as 40° refers to the -3 dB angle from zenith.**

What is the frequency resolution of NSMR and the Doppler profiles given in Figures 3 and 4?

**Changed Fig.3 caption to read "NSMR range-Doppler profiles for 19 July, 2020 (0.018 Hz frequency resolution), constructed from one minute observation periods."**

**Changed Fig. 4 caption to read: "Movement of a perturbation in profiles of the power at the strongest detection range in each frequency bin (0.018 Hz resolution) of PMSE detection for NSMR on 19 July, 2020."**

L146 – 150. In addition, the shape of the Doppler spectra changes, which is not discussed.

**Changed to read: "In this series of plots, it can be seen that a region of enhanced signal return moves from positive to negative Doppler, changing the shape of spectral power distribution over time."**

L220. What is the meaning of linear electron density with units el/m? Is it physically meaningful because even within the trail the electron density should be el/m^3 as it is 3 dimensional. This part needs better explanation.

**Linear electron density describes the total number of electron in a one meter thick slice perpendicular to the direction of meteor travel (i.e. integrated radially). It is a standard characterization of meteor trail plasma and has been used consistently at least as far back as the 1940s, as described by the cited references within this paragraph (McKinley 1961, Lovell et al. 1947). For clarity, changed to read: "Underdense meteor trails, with linear electron densities along the trail axis of less than..."**

Comparing Figures 7 and 1 is interesting. However, Figure 1 is only between 4 and 12 UT. Better give from 0 UT as there is a clear reduction in echo decay times around 1 UT.
Otherwise, including a new Figure with the observation for three days 18 – 20 July by both the radars will also be helpful here. In such a case, Figure 1 can be kept as such between 4 and 12 UT.

**We have elected to maintain the original plot ranges for the following reasons. There is no PMSE detection prior to 0500 or after 1300. This is clearly described in section 3. Maintaining the original time frames in Fig.1 and Fig.2 enhances visible details of PMSE and extending the time range in these plots would include RF interference, ionospheric return, and aircraft return that may confuse the reader. For Fig.7, we believe that showing the full 24 hours of meteor data helps acquaint the reader with the significant dynamical variability present in meteor decay times. An**

**extended analysis of 18-20 July is being included in an upcoming paper that will describe new analysis techniques.**

The Aspect sensitivity estimates are shown to match with previous measurements roughly. However, the following question arises in particular. Based on thin layer PMSE assumption and regions away from the zenith-wind vector plane fills the interior of the arc, for aspect sensitivity calculation, the thickness of the layer in all the Dopplers should be the same, right? However Figure 3 shows that the the thickness of negative Dopplers are less in the profile of 11:12 UT, for example. Does it represent the variabilities within the PMSE layers and in that case whether the aspect sensitivity measurements will not be affected?

**Added to the first paragraph of section 3.4: "The bottom of detected PMSE range-Doppler profiles is generally smooth and in good agreement with the calculated range-Doppler curve for a fixed height, which is indicative of a relatively flat bottom surface of the PMSE layer. The upper bound of PMSE return however exhibits significant variation, including differences in thickness and localized regions of enhanced return."**

**Variabilities within the PMSE layer affecting estimates of aspect sensitivity are addressed in the second paragraph of section 3.4 and are shown in figure 4.**

L268-273. Is it always that peak power observed at lowest range? Is there any criteria fixed on this in profile selection for aspect sensitivity calculations?

**Changed to read: "Hence, the peak powers observed in each frequency bin, which occur at heights that follow an arc closely parallel to the lower range boundary of PMSE return, provide an opportunity to translate observed Doppler shift into an estimate of zenith angle along the wind vector."**

Minor:
P1 L 20-22. It is well known that the gravity wave driven mesospheric circulation results in an adiabatic expansion-like situation in the summer polar region which reduces the temperatures to the extent facilitating formation of PMSE. It is misleading to mention that 5-day planetary waves may be responsible for temperature reduction based on one case study. It is likely that the 5-day planetary wave adds to the reduced temperatures and modulate PMSE formation by modulating the temperature.

**Changed to read: "…may contribute to low temperatures necessary to facilitate PMSE."**

How many profiles are integrated to get the 1 min time resolution for Doppler profiles?

**We are unclear on what reviewer 2 is asking here. The Doppler profiles are generated by applying a Fourier transform to 1-minute long time series. Added "…, generated from 1-minute long time series." to first paragraph of section 3.2. Hopefully this clarifies.**

Figure 2, y-axis should be extended to 120 km to make comparison easier with Figure 1. Lack of data at higher heights of SSR can be indicated with white color.

**Fixed.**

Provide a reference for eqn. 1.

**Equation 1 is derived from trigonometry and spherical geometry. As such, we have elected to leave equation 1 without a reference.**

L177. What range of shears are considered moderate? Is the data resolution 1.8 km here?

**Added after moderate description: "(as compared to observed values over the 24-hour period)"**

Figure 5. The start and end duration of main PMSE period may be highlighted with dashed vertical lines for easy reading and comparison.

**Line 161 clearly states that meteor winds are calculated at 2 km vertical and 30 minute temporal resolution. Meteor data do not have a fixed vertical resolution due to the scattered nature of meteor detections and the interplay between angular and range uncertainties. The method of calculating wind estimates from meteor drift is described in the cited reference.**

In section 5, L274, 278, 280. It is mentioned as 'zenith (frequency). Shouldn't it be zenith angle (frequency)?

**Changed to read "zenith angle"**

---

## Author Response (AR2)

The authors thank both reviewers for their efforts in reviewing the manuscript. In response to reviewer #2's suggestions, we have made the following changes, with author responses in bold:

Regarding the usage of the term linear electron densities: The authors have explained it in the response but in the revision they just mentioned '… with linear electron densities along the trail axis of..'. Is it along the axis or orthogonal to the axis? It is better to add a sentence in similar lines to what is given in the reply because Aeronomy community is not necessarily aware of the Meteor science terminology, though this is a technical paper using meteor radar.

**Added: "The durations of radar echoes from weakly ionized meteor trails, which constitute the overwhelming majority of meteor trail detections, provide information about the state of the background atmosphere in which they occur. The density of plasma in a meteor trail is usually characterized by the 'linear electron density', which is a measure of the radially integrated number of free electrons in a one-meter long segment of meteor trail (along the direction of meteoroid travel)."**

I do not understand their response to minor comment on figure 5 (presently figure 6). It was just a suggestion to include two vertical lines in the plots to indicate the start and end times of main PMSE duration studied here. This will help the reader to associate the shears during PMSE periods. The response given speaks about the meteor wind resolutions that is not the point made in the comment. However, I leave it to the author's decision as it is only a minor suggestion.

**Added dotted boxes to wind speed and sheer plots to denote PMSE detection periods on Fig. 6. Added to Fig. 6 caption: "Dotted boxes indicate PMSE detection periods. Gray squares denote insufficient data."**